# Agave Fructans in Oaxaca’s Emblematic Specimens: *Agave angustifolia* Haw. and *Agave potatorum* Zucc.

**DOI:** 10.3390/plants11141834

**Published:** 2022-07-13

**Authors:** Ruth E. Márquez-López, Patricia Araceli Santiago-García, Mercedes G. López

**Affiliations:** 1Instituto Politécnico Nacional, Centro Interdisciplinario de Investigación Para el Desarrollo Integral Regional—Unidad Oaxaca, Oaxaca 71230, Mexico; rmarquezl2100@ipn.com; 2Departamento de Biotecnología y Bioquímica, Centro de Investigación y de Estudios Avanzados del IPN—Unidad Irapuato, Guanajuato 36824, Mexico

**Keywords:** agave, fructans, age, degree of polymerization, FT-IR, HPAEC-PAD

## Abstract

Despite the recognition of *Agave tequilana* Weber var. Azul as raw material for producing tequila and obtaining prebiotics, there are other highly relevant *Agave* species in Mexico. Oaxaca contains a startlingly diverse range of *Agave* species; *Agave angustifolia* Haw. and *Agave potatorum* Zucc. are two classic specimens with great commercial potential. In this study, we examined the fructan fluctuation in these two species during their lifetime in the field (from 1 to 6 years old). First, we analyzed their morphological diversity based on vegetative characteristics. Subsequently, fructan extracts were analyzed by TLC, FT-IR, and HPAEC-PAD to identify carbohydrates. Multivariate analyses of the morphological parameters indicated a morphological divergence between the two species. Furthermore, we found that the concentration of simple carbohydrates and fructans, as well as the fructan DP, changed during plant development. Glucose, fructose, and fructooligosaccharides (FOS) were more abundant in *A. potatorum*, while *A. angustifolia* showed a greater amount of sucrose and fructans with a high DP. Fructan DP heatmaps were constructed using HPAEC-PAD profiles—the heatmaps were very helpful for establishing an easy correlation between age and the carbohydrate types present in the fructan extracts. This study is an important contribution to the agave fructan knowledge of the Mexican agave diversity.

## 1. Introduction

In most plants, sucrose and starch produced by photosynthesis are important carbon sources for skeletons and energy. Nevertheless, approximately 15% of flowering plant species store fructans as the main reserved carbohydrate [1]. Fructans are polymers formed by chains of fructose residues interconnected by glycosidic linkages. Different classes of fructans are distinguished according to the position of the sucrose moiety, the kind of linkage between the fructosyl units, and the degree of polymerization (DP) [2]. Inulin-type and levan-type fructans are linear molecules composed of linear β-D-fructose units linked by a β(2,1) and β(2,6) glycosidic bond, respectively [3]. Neoseries-type fructans are also linear molecules built up from neokestose and containing an internal glucose. Depending on the predominant glycosidic bond (β(2,1) or β(2,6)), fructans can be subdivided into the neo-inulin type and/or the neo-levan type. Graminan-type fructans, on the contrary, are more complex molecules, with a branched structure containing both β(2,1)- and β(2,6)-linked fructosyl-fructose units and with a terminal glucose residue [4]. Finally, agavins are so far the most complex fructans that have been found in different *Agave* species. These are composed of highly branched neo-fructans that include β(2,1) and β(2,6) linkages [5,6]. Depending on the number of fructose molecules (degree of polymerization), fructans are classified as fructooligosaccharides (FOS) if they have <10 fructose units and polysaccharides if they have >10 units.

The *Agave* genus is endemic to the Americas and comprises at least 206 species. Mexico is the primary center of diversity for this genus, as approximately 75% (159) of *Agave* species occur in its territory [7,8]. Agaves are economically important plants in Mexico because they are a source of many products, such as food, fiber, construction materials, and beverages, to mention some [7,9]. Although for a long time the highest market demand for agaves was for the distilled spirit production of tequila and mezcal, in recent years the agave demand has grown due to the appearance of two new agave byproducts: agave syrup [10], and as an important source of fructans [11]. 

Mancilla-Margalli and López [6] reported the structures of the agavins for different *Agave* species, including *A. angustifolia* and *A. potatorum*, which are species with ecological, agronomic, and economic relevance and are symbols of regional cultural identity in Oaxaca. However, they did not provide detailed information on the age of the agaves or the origin of the studied materials. *Agave angustifolia* Haw. is the most widely cultivated species in Oaxaca, while *A. potatorum* Zucc. stands out for being the raw material used to make Mezcal Tobalá, a Mexican alcoholic distillate highly appreciated for its mild sensory qualities. Recently, Soto et al. [12] reported that, regardless of age, *A. potatorum* accumulates bioactive compounds in its leaves, such as coumarins, cardiotonic glycosides, and fructans, which demonstrates the biotechnological potential of this species. In addition, León-Martínez and Ortíz-Hernández [13] observed that calcium oxalate crystals present in the epidermis of *A. potatorum* leaves reflect part of the incidence of light; their abundance was related to the evolutionary adaptation of this *Agave* species to drought stress conditions.

Recently, agave fructans have received considerable attention due to their prebiotic effect, since they selectively stimulate the growth and/or activity of potentially health-enhancing intestinal bacteria [14]. Additionally, agave fructans have technological properties that are closely determined by their composition, making them a suitable ingredient for improving many food products without affecting the sensory properties [15,16]. Despite the great diversity of agaves in Mexico, most investigations into fructan metabolism have focused on the *Agave tequilana* Weber Blue variety; however, much remains to be investigated for other *Agave* species. The aim of the current study was to extend the fructan metabolism knowledge of two of Oaxaca’s emblematic *Agave* species. In this study, we aimed to characterize the fructan extracts from two *Agave* species at different ages. We first performed a comparative analysis of morphological parameters. We then characterized fructans through a series of analytic techniques, including thin layer chromatography (TLC), Fourier-transform infrared spectroscopy (FT-IR), and high-performance anion-exchange chromatography with pulsed amperometric detection (HPAEC-PAD). 

## 2. Results

### 2.1. Morphology Analysis

For the measurements of morphological parameters, *Agave angustifolia* Haw. plants of 1 to 6 years of age were collected from San Esteban Amatlán, Oaxaca, Mexico, and *Agave potatorum* Zucc. plants of 1 to 6 years of age were collected from Infiernillo, Zaachila Oaxaca, Mexico (Figure 1). Ten morphological characteristics of *A. angustifolia* and *A. potatorum* were measured in randomly selected plants of different ages. We performed Shapiro–Wilk normality tests for all datasets, and based on these results we decided to carry out parametric analyses. The datasets were evaluated by Tukey’s multiple range test with the SAS 9.1 software (SAS Institute, Cary, NC, USA). Appendix A summarize the data for the measured characteristics. For all the characteristics, we found significant differences between the different ages of both species (*p* ≤ 0.05).

To investigate the differences and the degree of variation between the samples of three replications, an unsupervised principal component analysis (PCA) was used. Based on ten morphometric variables, the first two PCA components together explained 94.63% of the variance (PC1: 76.24%, PC2: 18.39%) (Figure 2). For the ordering of PC1, the variables that contributed significantly were plant length (PL), rosette diameter (RD), and the number of leaves (NL). In PC2, the most important variables were associated with the number of leaves (NL), leaf height (LH), and spine length (SL) (Appendix A). The PCA showed that the agaves were grouped according to species. Two large groups were observed. The ordination pattern showed the first group in a sector on the right (inside the pink circle), composed of individuals of the species *A. angustifolia*, which were classified by the length of the leaf (LL), the plant length (PL), and the rosette diameter (RD), with all characteristics defining the largest-sized plants. The second group is shown by the blue circle on the plot (Figure 2) and was composed of individuals of the species *A. potatorum*, which are smaller plants than those of the pink circle group, with shorter but wider leaves and a larger terminal spine.

### 2.2. Thin Layer Chromatography (TLC) Profiles of Fructans in A. angustifolia and A. potatorum Plants of Different Ages

The carbohydrate profiles of *A. angustifolia* Haw. and *A. potatorum* Zucc. plants (1−6-year-old plants) were identified by thin layer chromatography (TLC) (Figure 3). The fructan extracts were compared with a standard mixture (STD: G, glucose; F, fructose; S, sucrose; 1K, 1-kestose; 1N, nystose; and DP5), as well as commercial inulins of short (SDP) and high (HDP) degrees of polymerization (RSE, raftilose and RNE, raftiline, respectively). The carbohydrates were identified by their retention factors (Rf values) and spot colors: glucose (Rf = 0.81, bluish color), fructose (Rf = 0.81, reddish), sucrose (Rf = 0.75, brown), 1-kestose (DP3) (Rf = 0.5, brown), 1-nystose (DP4) (Rf = 0.43, brown), and 1-kestopentaose (DP5) (Rf = 0.37, brown). The color intensity was related to the carbohydrate’s type and abundance. TLC showed that the profiles of *A. angustifolia* and *A. potatorum* fructans (Figure 3) displayed a different profile when compared to RSE and RNE. This was because RSE is composed mainly of FOS of the Fn series (without a glucose unit) and RNE is composed mainly of GFn of the inulin fructan series, with a HDP, while agave fructans are a complex mixture of FOS and HDP fructans. Spots corresponding to glucose, fructose, and sucrose appeared in all samples, independently of the species. TLC analysis showed that 1−2-year-old *A. angustifolia* plants (Figure 3A) mainly stored SDP fructans, while the most abundant carbohydrates in 3−6-year-old plants were HDP fructans. The fructose/glucose spots were most intense in 1-year-old plants. TLC also showed that 1−3-year-old *A. potatorum* plants (Figure 3B) mainly stored SDP fructans, while the most abundant carbohydrates in 4−6-year-old plants were HDP fructans. Therefore, in general, fructans appear to have a higher DP in the adult plants of both species. 

### 2.3. Fourier-Transform Infrared (FT-IR) Spectroscopy of Fructans in A. angustifolia and A. potatorum Plants of Different Ages

FT-IR spectroscopy was used to evaluate the carbohydrate composition of the extracts. For comparative analysis purposes, the full spectra of different samples are depicted in Figure 4. Bands were identified by a comparative revision with the literature. FT-IR spectra of both plant species over the region 4000–600 cm^−1^ were collected. The spectra showed a strong and broad band in the 3000 to 2250 cm^−1^ regions, which is usually due to the stretching vibrations of hydroxyl (-OH) groups, including carbohydrates and phenolic hydroxyl groups [17]. On the other hand, there were also changes in the band at 3000–2800 cm^−1^ which corresponded to the symmetric and asymmetric stretching vibrations of skeletal CH and CH_2_ in polysaccharides [17]. In the 1500–1200 cm^−1^ spectral region, we observed bands attributable to local symmetric vibrations of CH_2_ and numerous C-OH deformations of carbohydrates [18]. However, the spectral area from 1800 to 800 cm^−1^ is a region with contributions from major chemical groups, and is frequently used in FT-IR analysis [19,20]. Strong absorption bands were exhibited in the 1300–900 cm^−1^ region. This region is attributed to C-C and C-O stretching and C-O-H and C-O-C bending, which are characteristics of several oligo- and polysaccharides [10]. The overlapped mode of the displayed spectra showed the resemblance of samples; however, differences among plants were observed in their carbohydrate and fructan contents (Figure 4). Chemometric analysis, using principal component analysis, was performed to determine whether there were any significant differences between the samples. The plants of both species were classified using only the specific carbohydrate (1300–900 cm^−1^) and fructan (950–920 cm^−1^) regions instead of the whole spectrum. The PCA model from the fingerprint regions coupled to carbohydrates and fructans in the same FT-IR spectra showed good separation by quadrants, making it possible to classify all samples using age and *Agave* species as criteria. Two specific sections were detected in the carbohydrate analysis. One was highly related to agave plant age, containing *A. angustifolia* of 1–6 years and *A. potatorum* of 6 years, with a greater concentration of carbohydrates reported in the positive values of component 1; on the other hand, *A. potatorum* plants of 1–5 years showed negative values (Figure 5A). Finally, the PCA for the fructan region displayed a similar behavior to that of the carbohydrate PCA model; two specific sections were detected. One was highly related to agave plant age, containing *A. angustifolia* of 3–6 years and *A. potatorum* of 5 and 6 years, with a greater concentration of fructans reported in the positive values of component 1 (Figure 5B). Component 2 may be related to the information about differences in the degree of polymerization. Based on the above, it is clear that fructans have a strong contribution to a sample’s classification based on its age.

### 2.4. Fructan Content According to Plant Age in A. angustifolia Haw. and A. potatorum Zucc.

To evaluate changes in fructan concentrations throughout the developmental cycle of *A. angustifolia* and *A. potatorum*, we determined the fructan content using the commercial “Fructans” (K-FRUC) analytical test kit (Megazyme International Ireland, Ltd., Wicklow, Ireland). Table 1 shows the fructan concentration changes in both species. The *A. angustifolia* fructan concentrations fell within 174.80−605.11 mg/g, while the *A. potatorum* fructan concentrations fluctuated between 115.65 and 502.11 mg/g. The lowest fructan concentration for both species occurred in 1-year-old plants (Table 1). After that age, a linear relationship between the increment in the fructan content and the plant age was observed. The highest concentration of fructans was recorded for 6-year-old plants. 

### 2.5. Identification of Fructans Using HPAEC-PAD of A. angustifolia and A. potatorum Plants of Different Ages

In this study, we analyzed the fructan profiles of *A. angustifolia* Haw. and *A. potatorum* Zucc. plants at different ages using HPAEC-PAD. To determine the fructan DP, the elution times of each peak obtained in the chromatographic profiles of the agave extracts were compared to the chromatographic profiles of RNE (Figure 6A,D). Low-molecular-weight fructans were predominant in 1-year-old *A. angustifolia* plants (Figure 6B). On the other hand, 6-year-old plants presented a considerable increase in high-molecular-weight fructans (Figure 6C). A similar behavior was observed for plants of *A. potatorum* (Figure 6E,F). The quantification of most of the above compounds from standards (Table 2 and Table 3) showed that young plants of both species mainly stored simple sugars as glucose, fructose, sucrose, and FOS. The quantification of FOS showed that the 1K in 1-year-old *A. potatorum* plants was five times more concentrated than that of 6-year-old of the same species, while the 1K in 1-year-old *A. angustifolia* plants was seven times more concentrated than the corresponding 6-year-old plants. The 1-nystose and DP5 were present in around two and eight times greater abundance, respectively, in 1-year-old *A. potatorum* plants than in 6-year-old plants, and a similar ratios were observed for 1-year-old *A. angustifolia* plants versus 6-year-old plants (Table 2 and Table 3).

### 2.6. Carbohydrate Profiling Analysis of Agave Plants at Different Ages

We analyzed samples of *A. angustifolia* and *A. potatorum* plants to assess the metabolic profiles as they relate to carbohydrates. To explore the carbohydrate metabolic changes in agave plants of different ages, their carbohydrate profiles were analyzed for specific differences. Hierarchical clustering and heatmaps of carbohydrate profiles helped to identify patterns in the metabolites that were enriched or depleted in samples of *A. angustifolia* and *A. potatorum* plants. The fold changes from the overall mean concentration for different ages are shown using color coding (Figure 7). The samples of *A. angustifolia* were grouped into two clusters: Cluster A, which had higher concentrations of simple carbohydrates (glucose, fructose, and sucrose) and FOS (kestose, nystose, and DP5), and Cluster B, which contained higher concentrations of FOS and fructans of a HDP (DP11–DP16) (Figure 7A). Furthermore, in observing the distribution patterns of the samples in the two clusters, the samples of 1-, 2-, and 3-year-old plants were mainly concentrated in Cluster A, and the samples of 4-, 5-, and 6-year-old plants were principally concentrated in Cluster B. This means that high metabolic differences existed between plants 1–3 years old and plants 4–6 years old. Similarly, two large clusters in *A. potatorum* plants were observed: Cluster A, which had higher concentrations of glucose, fructose, sucrose, and FOS (DP3–DP5), and Cluster B, which mainly contained greater concentrations of fructans of a HDP (DP11–DP16) (Figure 7B). In this case, samples of 1-, 2-, 3-, and 4-year-old plants were principally concentrated in Cluster A, and 5- and 6-year-old plants were concentrated in Cluster B.

## 3. Discussion

Carbohydrates are the most fundamental molecules for plant life because they are essential for the generation of energy and metabolic intermediates that are then used for the synthesis of macromolecules. Carbohydrates are broadly classified as simple carbohydrates (mono- and disaccharides) and complex carbohydrates, which includes polysaccharides (cellulose and starch) [21]. In addition, there is another type of carbohydrate known as fructans, which are fructose-based polysaccharides. Fructans in plants serve several physiological roles, mainly in biotic and/or abiotic stress resistance; thus, diverse research groups are increasingly interested in researching the fructan metabolism of different species. Agaves are monocotyledonous plants and members of the *Agavaceae* family that store fructans as their main carbohydrate source. To date, it is known that agaves produce more complex fructans than any other species, and with unique structures called agavins [5]. As the final products of cellular biochemical activity, fructans directly reflect agave qualities. As profound changes in carbohydrates occur during plant development, an understanding of these changes may help to facilitate agave breeding. In the present study, we determined changes in the metabolic carbohydrate profiles of *A. angustifolia* Haw. and *A. potatorum* Zucc., two economically important *Agave* species in the state of Oaxaca, Mexico.

### 3.1. Morphological Parameters

Although some *Agave* species can be recognized by their most important phenotypic characteristics, other species are difficult to differentiate [7]. For this reason, there have been various studies conducted on the genetic diversity in the *Agave* genus, using morphological characteristics to identify species through morphological and molecular markers [22,23]. For a better understanding of the morphological diversity between *A. angustifolia* Haw. and *A. potatorum* Zucc., we performed an analysis of morphological characteristics. Our results indicated that there was an evident morphological divergence between these two species. In general, *A. potatorum* showed lower values than those of *A. angustifolia* for all vegetative characteristics except for leaf width and apical spine length. This means that individuals of *A. potatorum* have a smaller size and they bear less leaves per plant. The morphological trends documented in this study are similar to those previously reported by [24]. In this study, the high percentage of the total variation explained by the first PC (76.24%) suggests that it contained variables that could discriminate between agave individuals from both species. The first PC was influenced mainly by the characteristics defining the plant size, including plant length, rosette diameter, the number of leaves, leaf length, and leaf width. These results are in accordance with the trends found in other *Agave* species used for producing fermented and distilled beverages, which confirms that an analysis of only vegetative characteristics is sufficient for distinguishing between *Agave* species [23,25]. On the other hand, the PCA also showed that the variables can clearly discriminate *A. angustifolia* individuals by their age, while *A. potatorum* plants were not so clearly defined. The high levels of diversity found in the *A. potatorum* plants of different ages might be due to the way this species reproduces—mainly through sexual reproduction. Seedlings produced from seeds are highly heterogeneous and show slow growth [26]. In contrast, sexual reproduction in *A. angustifolia* is practically nonexistent or very rare, since humans remove the inflorescences prior to harvest; therefore, *A. angustifolia* is propagated exclusively by vegetative means. This is probably the main reason why *A. angustifolia* individuals have greater homogeneity.

### 3.2. Thin Layer Chromatography (TLC)

TLC is a rapid method for separating and identifying compounds and is commonly used as an analytical tool to detect and characterize sugars [27]. The results of this chromatographic separation showed qualitative and quantitative differences between species and ages. The profiles of the fructans from the dried extracts of *A. potatorum* presented more intense spots and a greater number of FOS. This characteristic is relevant for evaluating *A. potatorum* as a better prebiotic candidate, because multiple beneficial attributes have been documented for FOS [28,29]. On the other hand, *A. angustifolia* showed different chromatographic profiles, since the spot belonging to the high degree of polymerization was more intense compared to *A. potatorum*. This suggests that the 1-FFT enzyme is much more active in this species from a very early age. Meanwhile, the presence of isomeric FOS was due to the activity of enzymes such as 6G-FFT and 6-SST at earlier stages in *A. potatorum*. TLC is a typical analytical tool; it is very preliminary, but very useful for evaluating or exploring the status of samples in a global manner. However, it should be complemented by other types of analytical techniques.

### 3.3. Fourier-Transform Infrared (FT-IR) Spectroscopy

To investigate the effect of the species and age of plants on the carbohydrate and fructan profiles, FT-IR spectroscopy combined with multivariate analysis was used as a first approach. FT-IR is a powerful tool with a wide field of applications ranging from the characterization of structural modifications due to biological processes to the identification of biomolecules. Furthermore, this analytical tool has many advantages, such as rapidity and simplicity without the demand for laborious sample preparation. Several studies have shown the applicability of FT-IR for the fast identification and quantification of mono- and disaccharides as well as fructans [30,31]. The FT-IR analyses were carried out using only the specific carbohydrate and fructan regions instead of the full IR spectrum. The region at 1300–900 cm^−1^ resulted from the stretching vibrations of C–O, C–C, ring structures, and the deformation of CH_2_ groups. These stretching vibrations mainly contribute to polysaccharides, and are therefore useful for fructan identification [32]. Recent studies have shown that FT-IR associated with PCA could provide very valuable information about the carbohydrate variation in agave plants [33]. For the carbohydrate region, scatter plots of the obtained scores with PC1 (which explained 83% of variability) × PC2 (which explained 15.8% of variability) revealed a visible group for both species. The scores of *A. angustifolia* were located on the positive side of PC1, and the *A. angustifolia* plants were characterized by containing the lowest content of simple sugars (Table 2, Figure 3A), while the scores of *A. potatorum* were on the negative side of PC1 except for the 6-year-old plants. *A. potatorum* plants showed a greater content of simple sugars (Table 3, Figure 3A). On the other hand, for the fructan region, the scores of 4–6-year-old plants were located on the positive side of PC1, while those of the younger plants were on the negative side of PC1. Briefly, the PCAs discriminated plants according to their species, age, and carbohydrate content. FT-IR could be considered a good, rapid, and easy method for obtaining an overview of the fructan composition in agave plants.

### 3.4. Enzymatic Assay and HPAEC-PAD

Although the TLC and FT-IR spectral analyses allowed us to identify metabolic trends in terms of some carbohydrate abundance, other techniques such enzymatic assays and HPAEC-PAD were essential for establishing qualitative as well as quantitative differences. The results obtained from the HPAEC-PAD demonstrated that the quality and type of fructans in *A. angustifolia* and *A. potatorum* were not only species-specific, but also age-dependent. Table 2 and Table 3 clearly show a decrease in the carbohydrate concentration (SC and FOS) based on the age of the plants. In the case of *A. angustifolia*, glucose was always the least abundant and decreased with age from 9.95 to 2.85 mg/mg. This was followed by fructose, which also decreased from 25.63 to 2.71 mg/mg. However, at the age of 6 years, both simple sugars were the same. Sucrose, on the other hand, reduced its amount by about half from age one to age six. Similarly, *A. potatorum* showed a decrease in carbohydrate levels with age; however, the levels of the glucose and fructose were higher, while the sucrose accumulation was greater. It was also very evident that the sucrose values fluctuated more in *A. angustifolia* than in *A. potatorum*. It is possible that the heterogeneity in sugar levels was due to the sucrose produced by photosynthesis being metabolized by vacuolar invertases. This theory is supported by the high amounts of glucose and fructose present in *A. potatorum*, while the large amounts of sucrose in *A. angustifolia* might be related to the fact that the plant must accumulate this sugar for “hijuelo” formation and development during its asexual reproduction [34], unlike *A. potatorum*, which is an *Agave* species with only sexual reproduction. On the other hand, there was not a large difference in the accumulation of FOS-type carbohydrates among species; however, in both cases the amounts of FOS also decreased with age. This is likely related to their use as a substrate for the biosynthesis of fructans of a high DP [35,36]. In the case of DP5, *A. angustifolia* and *A. potatorum* accumulated almost identical amounts. FOS biosynthesis has been widely documented in the literature by [37,38,39]. Finally, we believe that the age-based decrease in simple carbohydrates and FOS is closely related to an accumulation of fructans of a HDP. The lower amounts of simple carbohydrates and FOS in older age plants correlates with the observed greater amount of fructans of HDP.

### 3.5. HPAEC-PAD and Heatmaps

To establish more precise data on the changes in the carbohydrate content of agave species, we performed HPAEC-PAD analyses. Although there are reports on the variation in the content and composition of carbohydrates in some *Agave* species during their biological cycle, these reports have focused on the variation of simple carbohydrate (sucrose, glucose, and fructose) and some FOS (1-kestose, 6-kestose, neokestose, 1-nystose, and 1-fructofuranosylnystose) without a comprehensive report on precise DP changes. To better visualize the metabolic changes in the carbohydrates/fructans based on species and age, we generated conventional heatmaps using HPAEC-PAD profiles. The primary purpose of creating heatmaps combined with hierarchical cluster analyses (HCAs) was to find an easy correlation between age and the different carbohydrate types present in agave fructan extracts. Heatmaps are very useful because they emphasize different types of information about the metabolites analyzed in a sample—a carbohydrate sample, in this case. The information examined here included carbohydrate type, age, DP length, amount/quantity, and others. Heatmaps can also provide a correlation that infers and might help to facilitate the identification and/or classification of the involved carbohydrates of a specific age. Figure 7 shows heatmaps of 18-by-18 data matrices along with a hierarchical clustering dendrogram on the age axes, which were generated based on HPAEC-PAD carbohydrate profiles. In total, 16 different carbohydrates were identified (Figure 7). The different carbohydrates in each row can be visualized easily from left to right, highlighting interesting findings among the carbohydrates from young-age vs. old-age plants. The heatmaps showed that the concentrations of simple carbohydrates were greater in young plants than in older plants for both species. However, *A. angustifolia* (Figure 7A) presented a special feature because two main clusters were observed: one grouped plants of 1–4 years old and the other grouped plants from 5 to 6 years old. Similarly to other *Agave* species, the enrichment of simple sugars in very young plants suggests the strong activity of fructosyltransferases and energy metabolism related to their growth and further development [36]. This study confirms past findings that indicate there are greater concentrations of simple sugars in young than in adult plants of *Agave* spp. [31,33]. On the other hand, the presence of fructans of a HDP were featured in old plants (i.e., 5 and 6 years old), as shown in the far-right columns of Figure 7A. Mellado-Mojica and López [35] also reported a significantly greater fructan content in adult plants in comparison to young plants. Furthermore, Arrizon et al. [40] reported that 2-year-old plants exhibited the highest levels of free monosaccharides and low-molecular-weight fructans (DP3–DP6), while plants of 6 years old showed an increase in high-molecular-weight-fructans with a mean DP from 4 to 24. Unlike *A. angustifolia,* the heatmap of *Agave potatorum* (Figure 7B) showed fewer subclusters. One cluster grouped plants 1–3 years old and another grouped plants 4–6 years old. However, there was a similar trend for simple sugars (i.e., G, F, and S) and FOS (short fructans), which were very strongly associated with young-age plants. A transition from ages 3 to 5 from a short DP to a high DP was very clear. The concentration changes in carbohydrates for both species are consistent with previous studies in other *Agave* species, such *A. tequilana*. Substantial differences were highlighted for the accumulation of some carbohydrates, e.g., *A. angustifolia* presented a greater concentration of sucrose, while *A. potatorum* showed a greater accumulation of glucose and fructose (Table 2 and Table 3). It is possible that at least part of the differences were due to the requirement of *A. angustifolia* to accumulate sucrose for “hijuelo” formation and development during its asexual reproduction [34], while *A. potatorum* is an *Agave* species with only sexual reproduction. On the other hand, changes in fructan concentrations were more pronounced for *A. angustifolia* than for *A. potatorum*. Numerous studies have shown that fructan synthesis, utilization, and storage are dynamic processes that are strongly dependent on cell physiology, plant organs, environmental conditions, and the developmental plant stage [41,42,43]. Furthermore, we must not forget that the interactions between people and plants creates historical artificial selection, since farmers favor the reproduction of “good” phenotypes through different management practices. In this sense, *A. angustifolia* has a longer history of use, management, and human selection, whereas *A. potatorum* has only a recent history of cultivation of likely no more than 15 year [44]. In both species, traditional farmers have generated variations that are subject to human selection because people select the most vigorous plantlets in seed beds and nurseries for their plantations. Although plant size is apparently advantageous for people in terms of amount, because larger plants yield higher usable matter for producing mezcal, a large-sized plant producing a lower amount of fructans is not necessarily a good resource. It will be of great interest for future studies to characterize the fructans of both cultivated and wild plants, because the quality (not only the quantity) of the product is very important. Agave fructan complexity continues to be explored; therefore, there is a growing need for a better understanding of the relationship between age and, not only the amount of fructans, but also the type of fructans. The utility of heatmaps on this type of data is an enhancement to the fructan area research, and we have explored its potential utility in the current study. Finally, this approach maximizes the display of highly variable information that can be obtained from a single HPAEC-PAD run. We really believe that this approach will be applied in future work related to fructan research.

## 4. Materials and Methods

### 4.1. Plant Material

*Agave angustifolia* Haw. plants of 1 to 6 years (three each) of age were selected and collected from San Esteban Amatlán, Oaxaca, Mexico. Amatlán is located at the geographic coordinates 16°23′ N and 96°30′ W, 1500 m above mean sea level, with a temperature from 8 to 28 °C and a semi-dry subhumid climate with rain in the summer. Precipitation varies between 600 and 800 mm. The age of a plant corresponded to its number of years in the field, starting from the planting of the “hijuelo” (plant shoot). After plant collection, pines were cut into small sections according to the methodology proposed by Arrizon et al. [40]. Agave samples were stored in plastic bags at −20 °C until use.

*Agave potatorum* Zucc. plants of 1 to 6 years (three each) of age were selected and collected from Infiernillo, Zaachila Oaxaca, Mexico. Infiernillo is located at the geographic coordinates 16°89′ N and 97°19′ W, 1969 m above mean sea level, with a temperature from 10 to 26 °C and a semi-warm subhumid climate. Precipitation varies between 300 and 1200 mm. The age of a plant corresponded to its number of years in the field, starting from the planting of the seed. After plant collection, pines were cut into small sections according to the methodology proposed by Arrizon et al. [40]. Agave samples were stored in plastic bags at −20 °C until use.

### 4.2. Fructan Extraction

Fructans were extracted according to the protocol of Mellado-Mojica and López [35] with some modifications. Briefly, 100 g of agave fiber was extracted in 100 mL of 80% ethanol with continuous stirring for 1 h at 60 °C. The sample was filtered with cellulose filters (Whatman, grade 4, Maidstone, Kent, England) to eliminate the solid fraction, and the plant material was re-extracted twice using 100 mL of water for 30 min at 60 °C. The supernatants were mixed; chloroform was used to eliminate the organic material from the extract. The aqueous phase was concentrated on a vacuum rotatory evaporator (Büchi Rotovapor R-200, Brinkmann Instruments Inc., Westbury, NY, USA) at 60 °C. A spray dryer (Yamato, ADL 3115, Tokyo, Japan) was used to dry the samples. The sample was fed into the drying chamber using a peristaltic pump at a flow rate of 3 mL/min, and an inlet drying temperature of 140 °C was selected as the right temperature that permitted the recovery of a dried product. The outlet temperature of the exhaust air was 80 °C; the atomization pressure was 0.25 MPa. Samples were stored in a desiccator until further analyses.

### 4.3. Thin Layer Chromatography (TLC) 

A total of 1 μL of agave fructan solution (25 mg/mL) was loaded onto a pre-coated silica gel TLC plate, in addition to three standard mixtures at 25 mg mL^−1^; one composed of G, F, S, 1-kestotriose (DP3), 1,1-kestotetraose (DP4), and 1,1,1-kestopentaose (DP5), as well as Raftiline (RNE) and Raftilose (RSE). The plates were developed with a solvent system consisting of butanol/propanol/water [3:12:4 *v*/*v*/*v*] [45]. The developed plates were stained by spraying with a diphenylamine-aniline-phosphoric acid solution and heating at 150 °C for 10 s to achieve fructan visualization [46].

### 4.4. Fourier-Transform Infrared (FT-IR) Spectroscopy

Fourier-transformed mid-infrared spectra were obtained using a Cary 600 FT-IR spectrometer (Agilent Technologies, Folsom, CA, USA) equipped with an attenuated total reflectance (ATR) accessory and a diamond/Ge crystal plate (Agilent Technologies, Folsom, CA, USA) (MIRacle by PIKE Technologies, USA). For measurements, 50 mg of each sample was placed onto the ATR crystal plate and 32 scans were recorded in the range of 4000–600 cm^−1^ at a nominal resolution of 4 cm^−1^ in transmittance mode (%T). Single-beam spectra of all samples were collected using air as the reference background. After each measurement, the crystal was cleaned with pure ethyl alcohol and dried using a paper towel. In order to make sure the ATR crystal was clean; a spectrum was collected using the latest background as the reference. Three replicate measurements of each sample were taken and the spectra were averaged. Principal component analysis (PCA) was performed in the Pirouette software. Principal component models were developed for different regions of the FT-IR: carbohydrates (1300–900 cm^−1^) and fructans (959–920 cm^−1^).

### 4.5. Enzymatic Determination of Total Fructan Content in Extracts of A. angustifolia Haw. and A. potatorum Zucc. Plants

Total fructan quantification was performed with the commercial “Fructans” (K-FRUC) analytical test kit (Megazyme International Ireland, Ltd., Wicklow, Ireland) according to the instructions of the manufacturer. Samples were analyzed at a wavelength of 340 nm in a spectrophotometer (Varian Cary 50 Bio, UV-Vis, Walnut Creek, CA, USA). All measurements were carried out in triplicate.

### 4.6. Fructan Determination

Fructan concentrations were analyzed using HPAEC-PAD in the ion chromatograph Dionex ICS-3000 (Thermo Scientific, Sunnyvale, CA, USA), with the guard-column CarboPac PA-100 (4 × 50 mm) and a CarboPac-PA100 (4 × 250 mm) column. Fructan separation was achieved using the running conditions described by Mellado-Mojica and López [35]. The applied potentials, for detection by the amperometric pulse, were as follows: E1 (400 ms), E2 (20 ms), E3 (20 ms), and E4 (60 ms) of +0.1, −2.0, +0.6, and −0.1 V, respectively. Prior to their injection, the samples were diluted with deionized water (resistivity of 17 MΩ) to a concentration of 0.5 mg/mL and filtered through 0.45 μm (Millipore^®^, Burlington, MA, USA) nylon membranes. A total of 25 μL of diluted sample was injected into the HPAEC. Glucose, fructose, sucrose, 1-ketose, nystose, and fructosyl-nystose (Sigma Aldrich, St. Louis, MO, USA) were used as standards for quantification using linear regression analyses. Additionally, from the distribution profile of the Raftiline (RNE) from Beneo-Orafti (Tienen, Belguim), we estimated the degree of polymerization. The obtained results were expressed in nC versus time.

### 4.7. Data Processing and Multivariate Data Analysis

All results were analyzed using the R software program for statistics and graphical presentation (https://www.r-project.org; accessed on 1 March 2022) on Chemometrics of the morphological parameters were performed using unsupervised principal component analysis (PCA) in the R software environment. The differences of measured variables were evaluated by Tukey’s multiple range test with the SAS 9.1 software (SAS Institute, Cary, NC, USA). Differences were considered significant at *p* < 0.05.

The baseline correction and obtained data matrix of each FT-IR spectrum was done using Agilent Resolutions Pro 5.3 software. The data set was reduced by taking an average of the technical replicates, resulting in a data set with 32 spectra that were used in the data analyses. For the analyses of the FT-IR spectral set of samples, spectral regions of 1300–900 cm^−1^ and 950–920 cm^−1^ were selected, as these spectral regions contain distinctive bands for carbohydrates and fructans. Chemical similarities between samples were estimated by using principal component analysis (PCA). Pirouette 4.5 software was used for data import, pretreatment, and construction of the chemometric classification models.

Data acquisition in the HPAEC-PAD and the automatic integration of each chromatogram were done using the Dionex Chromeleon 6.8 software. The data were selected by peak, picking only chromatographic peaks, including mono- and disaccharides and FOS from DP3 to DP16. For the analysis of metabolic changes in different samples, heatmaps were carried out using the R software. The area of each peak was used as a single variable, producing a matrix of 16 variables. The raw data (non-normalized) was used for multivariate data analysis. The peak areas were converted to Z-scores. The row Z-score values were calculated as mean abundance subtracted from the abundance and then divided by the standard deviation across all samples. Furthermore, to obtain information about the similarity between samples, a hierarchical clustering analysis (HCA) with Euclidean distance and Ward’s algorithms was performed.

## 5. Conclusions

In the present study, we proposed several tools to facilitate carbohydrate research in *Agave* species of different ages. FT-IR can be used to easily establish agave fructan profiles to correlate with age, while HPAEC-PAD heatmaps may be very handy for explaining the DP behavioral patterns related to a metabolic question. These results, along with TLC, showed that the concentration of simple carbohydrates was greater in young plants and fructans of a HDP predominated in old plants for both species. However, the type and concentration of simple carbohydrates and fructans varied depending on the species. *A. angustifolia* Haw. presented greater concentrations of sucrose, while *A. potatorum* Zucc. showed a greater accumulation of glucose and fructose. On the other hand, *A. potatorum* was enriched predominantly in short- and middle-DP fructans. Meanwhile, *A. angustifolia* presented large amounts of fructans of a high DP. Furthermore, multivariate analyses suggested that fructans could be used as potential biomarkers for classifying agave plants, using age and species as criteria. Overall, this study also aimed to improve the overall understanding of the carbohydrate metabolism of agave plants. The detailed carbohydrate data presented here reveal information related to the timing of important switches in primary carbohydrate and fructan metabolism, which could provide some clues about the transcriptional and enzymatic activity for future research.

## Figures and Tables

**Figure 1 plants-11-01834-f001:**
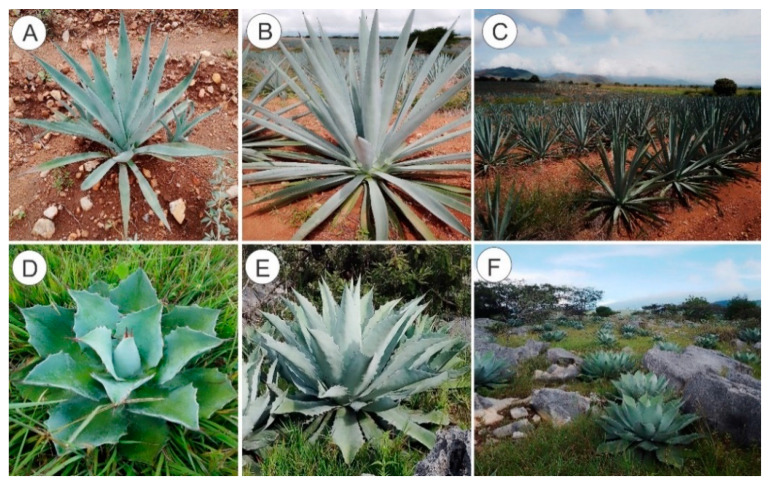
(**A**) *A. angustifolia* plant of one year. (**B**) *A. angustifolia* plant of six years. (**C**) *A. angustifolia* Haw. plants in San Esteban Amatlán, Oaxaca, Mexico. (**D**) *A. potatorum* plant of one year. (**E**) *A. potatorum* plant of six years. (**F**) *A. potatorum* Zucc. plants in Infiernillo, Zaachila Oaxaca, Mexico.

**Figure 2 plants-11-01834-f002:**
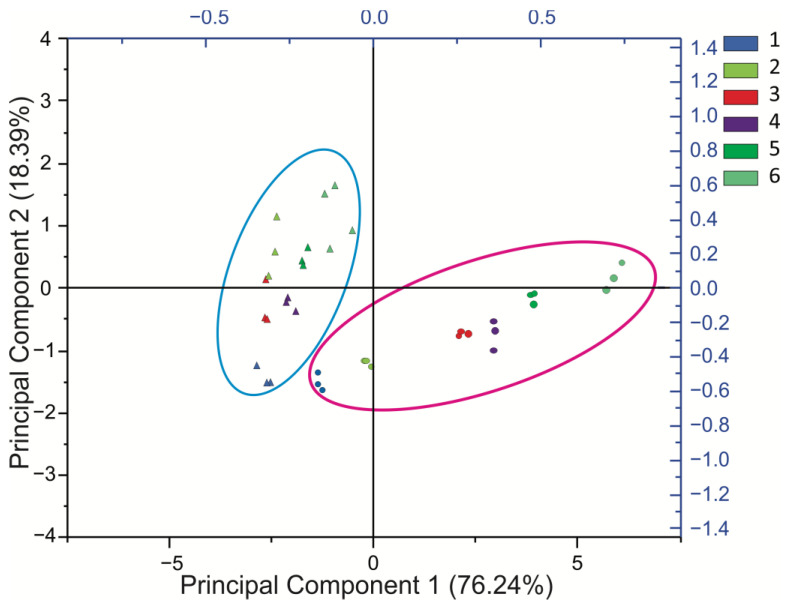
Principal component analysis (PCA) scatter plot of the first two coordinates based on ten morphological characteristics measured in plants of *A. angustifolia* and *A. potatorum* of different ages. Circles within the pink oval indicate *A. angustifolia* plants, while triangles within the blue oval indicate *A. potatorum* plants. The numbers and colors outside the graph indicate the age of the plants.

**Figure 3 plants-11-01834-f003:**
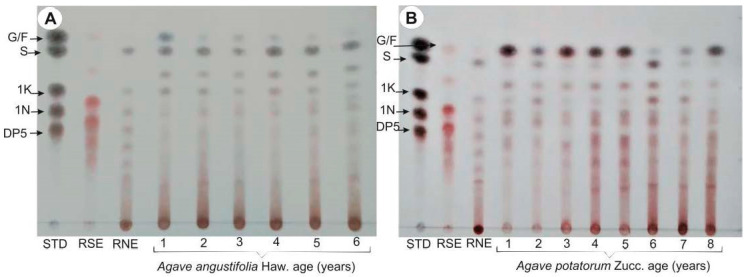
Thin layer chromatography (TLC) from (**A**) *A. angustifolia* and (**B**) *A. potatorum* plants at different ages. The standard nomenclature is as follows: STD, standards; RSE, raftilose; RNE, raftiline. Mobile phase: butanol/propanol/water; stationary phase: silica; derivatizing reagent: diphenylamine-aniline-phosphoric acid. See other nomenclature details in the Materials and Methods section.

**Figure 4 plants-11-01834-f004:**
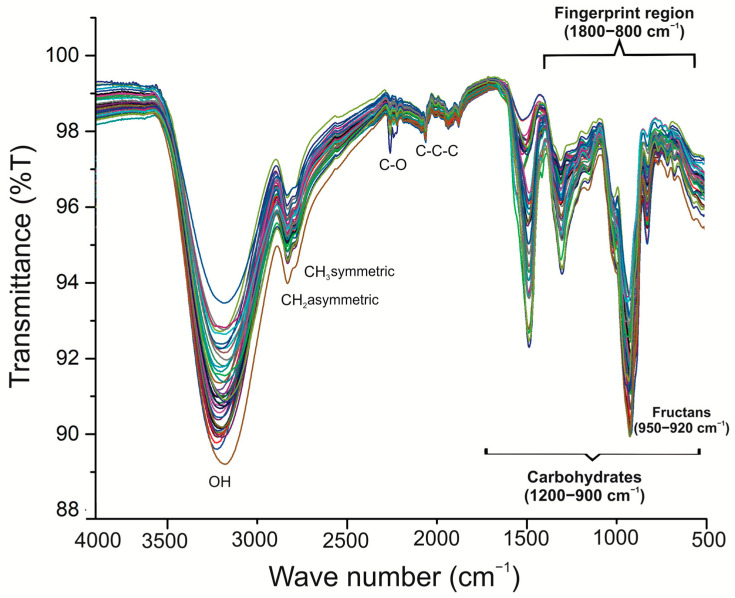
FT-IR full spectra of the agave fructans from *A. angustifolia* Haw. and *A. potatorum* Zucc. plants of different ages.

**Figure 5 plants-11-01834-f005:**
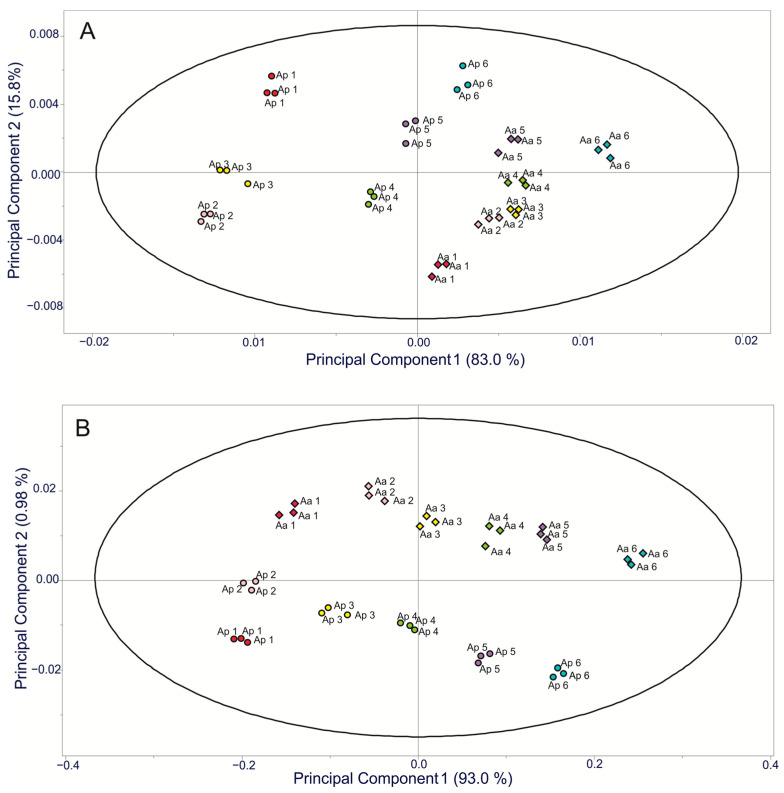
Principal component analysis (PCA) models for FT-IR spectra of agave fructans from *A. angustifolia* and *A. potatorum* plants of different ages. (**A**) Carbohydrate region (1300–900 cm^−1^); (**B**) fructan region (950–920 cm^−1^).

**Figure 6 plants-11-01834-f006:**
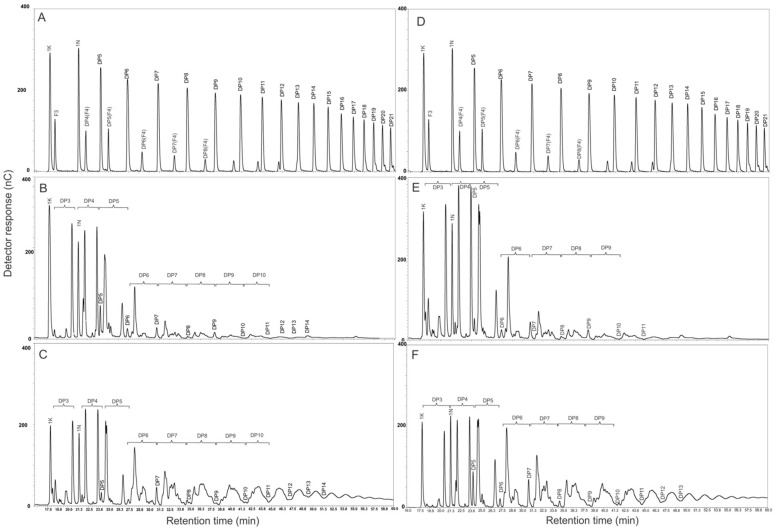
HPAEC-PAD profiles of fructans from (**A**,**D**) raftiline (RNE), (**B**) 1-year-old *A. angustifolia* plants, (**C**) 6-year-old *A. angustifolia* plants, (**E**) 1-year-old *A. potatorum* plants, and (**F**) 6-year-old *A. potatorum* plants.

**Figure 7 plants-11-01834-f007:**
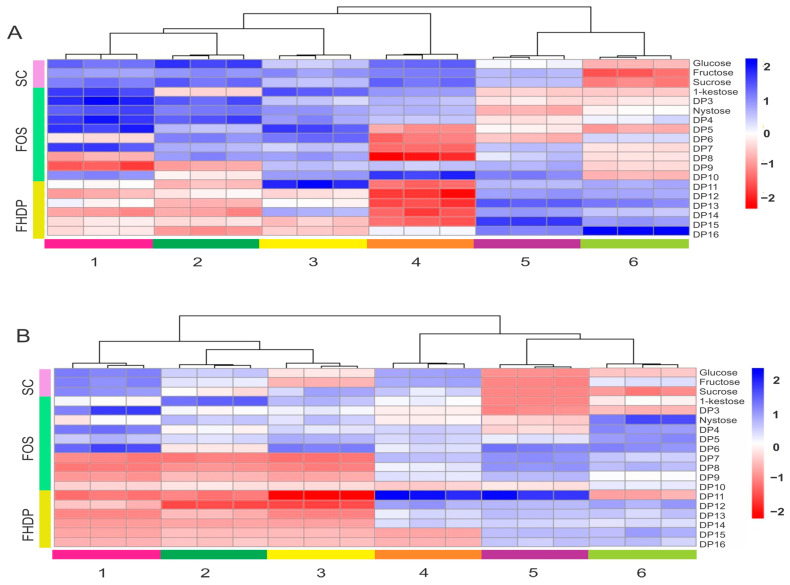
Hierarchical clustering analyses (HCA) and heatmaps of carbohydrate concentrations of agave plants at different ages. (**A**) *Agave angustifolia;* (**B**) *Agave potatorum*. Metabolites (carbohydrates) are represented by each row and listed on the right side of the heatmap, while each sample and its biological replicates are represented by columns. All peak areas were normalized to Z-scores for each carbohydrate. The color scale represents fold changes with respect to the average value of each metabolite, with the white color representing an average relative abundance, the blue color indicating a higher metabolite concentration than the median of the samples, and the red color indicating a lower concentration, as shown to the right of the color bar. Abbreviations: SC, simple carbohydrates; FOS, fructooligosaccharides; FHDP, fructans with a high degree of polymerization.

**Table 1 plants-11-01834-t001:** Fructan content of *A. angustifolia* Haw. and *A. potatorum* Zucc. at different ages.

Age (Years)	*A. angustifolia* (mg/g)	*A. potatorum* (mg/g)
**1**	174.80 ± 2.29	115.65 ± 2.89
**2**	252.87 ± 3.02	128.67 ± 2.06
**3**	321.61 ± 2.18	251.65 ± 2.43
**4**	434.50 ± 3.62	344.60 ± 3.32
**5**	564.84 ± 3.97	464.84 ± 3.97
**6**	605.11 ± 3.71	502.11 ± 3.71

**Table 2 plants-11-01834-t002:** Carbohydrate and fructooligosaccharide content of *A. angustifolia* Haw. plants at different ages.

Age (Years)	1	2	3	4	5	6
	**Carbohydrate Contents ***					
**Glucose**	9.95 ± 1.64	8.31 ± 0.71	5.77 ± 1.53	5.12 ± 1.07	5.43 ± 0.78	2.85 ± 0.90
**Fructose**	25.63 ± 0.68	14.61 ± 3.10	11.41 ± 0.92	6.85 ± 0.91	3.98 ± 0.68	2.71 ± 0.85
**Sucrose**	42.65 ± 1.76	35.74 ± 2.23	35.06 ± 1.86	32.31 ± 1.27	26.95 ± 1.38	22.76 ± 2.20
	**Fructooligosaccharide Contents ***					
**1-kestose (1K)**	17.01 ± 2.39	15.08 ± 1.61	15.55 ± 2.84	8.12 ± 0.87	1.42 ± 0.13	2.24 ±0.07
**1-nystose (1N)**	10.66 ± 1.39	10.54 ± 0.91	10.00 ± 0.48	8.51 ± 0.16	7.81 ± 0.32	6.66 ± 1.56
**1-kestopentaose (DP5)**	8.92 ± 1.03	6.62 ± 1.13	3.21 ± 0.28	0.92 ± 0.04	0.38 ± 0.06	1.31 ± 0.05

* mg/g of fructans (dry weight).

**Table 3 plants-11-01834-t003:** Carbohydrate and fructooligosaccharide contents of *A. potatorum* Zucc. plants at different ages.

Age (Years)	1	2	3	4	5	6
	**Carbohydrate Contents ***					
**Glucose**	40.38 ± 2.04	35.00 ± 1.82	31.55 ± 1.41	28.40 ± 1.86	22.72 ± 1.79	26.10 ± 0.17
**Fructose**	46.49 ± 1.84	38.32 ± 1.26	37.22 ± 1.60	37.22 ± 2.90	16.96 ± 2.86	25.33 ± 0.23
**Sucrose**	30.00 ± 2.71	19.26 ± 1.27	15.94 ± 3.18	8.25 ± 1.05	7.00 ± 1.72	6.29 ± 0.35
	**Fructooligosaccharide Contents ***					
**1-kestose (1K)**	29.57 ± 1.05	24.26 ± 1.83	25.09 ± 1.37	10.43 ± 1.79	8.27 ± 0.33	8.64 ± 1.28
**1-nystose (1N)**	16.38 ± 2.11	16.20 ± 0.68	13.99 ± 3.06	10.06 ± 1.90	10.64 ± 0.36	9.77 ± 0.33
**1-kestopentaose (DP5)**	8.79 ± 1.99	6.47 ± 0.93	5.41 ± 0.08	4.55 ± 0.28	3.55 ± 0.28	2.73 ± 0.56

* mg/g of fructans (dry weight).

## Data Availability

Not applicable.

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
