# Peer review of "Agave Fructans in Oaxaca’s Emblematic Specimens: Agave angustifolia Haw. and Agave potatorum Zucc."

_plants, 2022, doi:10.3390/plants11141834_

Round 1

Reviewer 1 Report

The paper of Márquez-López et al. deals with the assessment of the carbohydrate metabolic changes in Agave plants of different ages. In particular, two economically important Agave species in the state of Oaxaca, Mexico, were analyzed by means of morphological diversity and fructans fluctuations characterized through a series of analytical techniques such as TLC, FT-IR, and HPAEC-PAD. The important part of analyses is performed on the base of chemometric methods. In general, the issue of the paper is interesting and the manuscript is well presented.

However, I have some comments/suggestions/questions:

  • there is no clear conclusion regarding the obtained results;
  • is possible to explain even in more detail the performed data processing and multivariate data analysis;
  • what can you say about the other spectral changes observed in the recorded FT-IR spectra; I really miss some deeper interpretation of the spectra;
  • please, take care of the organization of the figures and table throughout the text of the manuscript - there is a lot of empty space;
  • please, take care about the typing errors, especially, spaces; this is also the case with the list of references (there is no need to add a line after each reference).

Author Response

Reviewer # 1

The paper of Márquez-López et al. deals with the assessment of the carbohydrate metabolic changes in Agave plants of different ages. In particular, two economically important Agave species in the state of Oaxaca, Mexico, were analyzed by means of morphological diversity and fructans fluctuations characterized through a series of analytical techniques such as TLC, FT-IR, and HPAEC-PAD. The important part of analyses is performed on the base of chemometric methods. In general, the issue of the paper is interesting, and the manuscript is well presented.

However, I have some comments/suggestions/questions:

ANSWERS

1.- There is no clear conclusion regarding the obtained results.

Answer: We are very grateful to the reviewer; we have taken it into account. We have proceeded to change the conclusion (lines 536-552).

2.- Is possible to explain even in more detail the performed data processing and multivariate data analysis?

Answer: We agree with the reviewer. We have added information of multivariate analyses in the data processing and multivariate data analysis section (line 410-435).

3.- What can you say about the other spectral changes observed in the recorded FT-IR spectra; I really miss some deeper interpretation of the spectra.

Answer: Thank you for your suggestion. We have worked on the manuscript and added the interpretation of the FT-IR spectra in the results section (lines 145-155) and figure 4.

4.- Please, take care of the organization of the figures and table throughout the text of the manuscript - there is a lot of empty space.

Answer: Thanks for the suggestion, we have corrected the organization of the figures and tables.

5.- Please, take care about the typing errors, especially, spaces; this is also the case with the list of references (there is no need to add a line after each reference)

Answer: Thanks for the observation, we have checked and corrected the typing errors in the manuscript.

NOTE: All changed were highlighted in yellow in the new manuscript version.

Reviewer 2 Report

Agave fructans are natural reserve carbohydrates in agave plants. Today, agave fructans have begun to be used as functional ingredients in various food products due to their favorable impact on health. The Agave genus includes at least 206 species, but despite the great diversity of agaves, most investigations have focused on the Agave tequilana. Therefore, much remains to be done with other Agave species and this article expands the knowledge on this subject.

The topic falls in the scope of this journal. I found this article complete, detailed and well written. The design is reasonable and the authors have given a good number of citations about the subject. Therefore, I consider the manuscript suitable for publication.

I recommend making some minor revisions before publishing the manuscript:

1)    lines 53, 60, 63, 390, 392, 438, 446, 449, etc. - change the references within the text, removing the brackets with the year and inserting the square brackets and the reference number, e.g. “Mancilla and López (2006)” with Mancilla and López [6] or “Soto et al., 2021” with Soto et al. [12]

2)    line 171 – Figure 4, correct the name of the y-axis (Transmittance) and delete a dot at the end of the sentence

3)    line 451 - Which filter was used?

4)    line 454 – “evaporation and spray-dried” I recommend detailing the instruments used

5)    line 461 – “butanol/propanol/water” in which ratio?

6)  line 509 - change paragraph numbering (5).

Author Response

Reviewer # 2

Agave fructans are natural reserve carbohydrates in agave plants. Today, agave fructans have begun to be used as functional ingredients in various food products due to their favorable impact on health. The Agave genus includes at least 206 species, but despite the great diversity of agaves, most investigations have focused on the Agave tequilana. Therefore, much remains to be done with other Agave species and this article expands the knowledge on this subject.

The topic falls in the scope of this journal. I found this article complete, detailed and well written. The design is reasonable, and the authors have given a good number of citations about the subject. Therefore, I consider the manuscript suitable for publication.

I recommend making some minor revisions before publishing the manuscript:

ANSWERS

1.- Lines 53, 60, 63, 390, 392, 438, 446, 449, etc. - change the references within the text, removing the brackets with the year and inserting the square brackets and the reference number, e.g. “Mancilla and López (2006)” with Mancilla-Margalli and López [6] or “Soto et al., 2021” with Soto et al. [12].

Answer: We appreciate your comment. We have made the recommended changes (lines 53, 60, 63, 390, 392, 438, 446, 449, etc.).

2.- Line 171 – Figure 4, correct the name of the y-axis (Transmittance) and delete the dot at the end of the sentence.

Answer: Thanks for the observation, we have corrected these mistakes.

3.- Line 451 - Which filter was used?

Answer: Thanks for the observation, the information was introduced in the manuscript in materials and methods section (line 456-457).

4.- Line 454 - “evaporation and spray-dried” I recommend detailing the instruments used.

Answer: We appreciate your comment and have introduced this information in materials and methods section (lines 459-466).

5.- Line 461 - “butanol/propanol/water” in which ratio?

Answer: We appreciate your observation and have added the corresponding information to materials and methods (line 472).

6.- Line 509 - change paragraph numbering (5).

Answer: We appreciate your valuable comment and have corrected this mistake (line 536).

NOTE: All changed were highlighted in yellow in the new manuscript version.
